# Trends in Exclusive, Dual and Polytobacco Use among U.S. Adults, 2014–2019: Results from Two Nationally Representative Surveys

**DOI:** 10.3390/ijerph182413092

**Published:** 2021-12-11

**Authors:** Delvon T. Mattingly, Luis Zavala-Arciniega, Jana L. Hirschtick, Rafael Meza, David T. Levy, Nancy L. Fleischer

**Affiliations:** 1Department of Epidemiology, School of Public Health, University of Michigan, Ann Arbor, MI 48109, USA; lzavalaa@umich.edu (L.Z.-A.); janahirs@umich.edu (J.L.H.); rmeza@umich.edu (R.M.); nancyfl@umich.edu (N.L.F.); 2School of Medicine, Georgetown University, Washington, DC 20057, USA; dl777@georgetown.edu

**Keywords:** cigarettes, electronic nicotine delivery systems, cigars, pipes, smokeless tobacco, dual use, polyuse, tobacco products, prevalence

## Abstract

Although increases in the variety of tobacco products available to consumers have led to investigations of dual/polytobacco use patterns, few studies have documented trends in these patterns over time. We used data from the 2014/2015 and 2018/2019 Tobacco Use Supplement to the Current Population Survey (TUS-CPS) and the 2015–2019 National Health Interview Survey (NHIS) to estimate trends in the following use patterns: exclusive use of cigarettes, electronic nicotine delivery systems (ENDS), other combustibles (cigars/cigarillos/little filtered cigars and traditional pipes/hookah), and smokeless tobacco (four categories); dual use (two product groups) of each product group with cigarettes (three categories); polyuse with cigarettes (all four product groups; one category); and dual/polyuse without cigarettes (one category). We estimated trends in product use patterns overall and by age, sex, and race/ethnicity using two-sample tests for differences in linear proportions. From 2014/2015 to 2018/2019, exclusive ENDS use increased, whereas cigarettes and ENDS dual use decreased. Furthermore, polyuse with cigarettes decreased, whereas dual/polyuse without cigarettes increased, with trends varying by age, sex, and race/ethnicity. Our findings suggest that patterns of dual/polyuse with and without cigarettes have changed in recent years, indicating the need for further surveillance of concurrent tobacco product use patterns.

## 1. Introduction

Tobacco use is the leading cause of preventable death in the United States (USA) [1]. The tobacco industry is continuing to introduce new products to the market; therefore, the impact on population health remains a significant concern [1,2]. Monitoring tobacco product use, including dual use (use of two tobacco products) and polyuse (use of three or more tobacco products), is critical to gauge trends in product use patterns and project the health consequences of these trends [3].

Research on patterns of multiple tobacco product use in U.S. adults has increased over the past decade [4,5]. In a systematic review of dual use and polyuse, eleven studies have presented prevalence estimates of dual use, and six studies have presented estimates of polyuse from 1992 to 2018 using data from nationally representative studies [5]. This review found that dual use and polyuse are becoming more common throughout the world, and that definitions of dual use and polyuse use have varied over time. Dual/polyuse is also higher among younger adults compared to older adults in the United States [5,6]. In addition, results from a recent study analyzing three nationally representative surveys for the years 2014–2016 reported that the most common adult dual use and polyuse groups in the United States were cigarettes and electronic nicotine delivery systems (ENDS) dual use (1.3–1.8%), and cigarettes, ENDS, and other combustibles (OC, i.e., cigars/cigarillos/little filtered cigars and traditional pipe/hookah) polyuse (0.2–0.7%) [4]. 

Although dual use and polyuse trends may not be clear in recent years in the United States, trends in individual tobacco product use among adult populations are well documented. For example, studies have shown that cigarette use prevalence is decreasing among youth [7] and adults [8,9]. Furthermore, results from a nationally representative survey suggest that adult ENDS use may be increasing over time, particularly among young adults [10,11], corroborating prior research that found rapid growth in ENDS use in the early 2010s [12]. One study using data from the National Health Interview Survey (NHIS) reported that the contemporary prevalence of cigar use remained relatively stable from 2005 to 2015 (2.3%) [13], although another study using data from the National Surveys on Drug Use and Health (NSDUH) reported that contemporary prevalence of cigar use decreased from 2002 (7.9%) to 2016 (6.3%) [14]. Studies using the Tobacco Use Supplement to the Current Population Survey (TUS-CPS) and the NSDUH reported that adult smokeless tobacco use (SLT) prevalence has changed little in recent years [15,16].

Many studies that have explored dual use or polyuse trends did not include a broad range of tobacco products, instead examining the dual use of cigarettes and e-cigarettes [17,18,19], dual use of cigarettes and cigars or SLT [20,21], or polyuse without ENDS [22]. Moreover, studies have not characterized trends by sociodemographic group, precluding the identification of those most susceptible to the consequences of using multiple tobacco products, including a higher frequency of use, lower likelihood of quitting, and higher risk of both nicotine dependence and poor health outcomes than exclusive tobacco use [23,24,25]. Tobacco use patterns are evolving rapidly; therefore, further investigation using more recent data is needed. This study examined trends in exclusive, dual, and polyuse overall and by age, sex, and race/ethnicity, using 2014–2019 data from two nationally representative U.S. surveys: NHIS, a survey used to examine health trends in the United States, and TUS-CPS, the largest survey on tobacco product use in the United States.

## 2. Materials and Methods

### 2.1. Design

We analyzed data from the 2014/2015 and 2018/2019 TUS-CPS and 2015–2019 annual NHIS obtained from the Integrated Public Use Microdata Series (IPUMS) [26].

TUS-CPS is a cross-sectional survey of tobacco use behaviors among adult respondents or proxies aged 18 years or older administered every 3–4 years by the U.S. Census Bureau. The 2014/2015 TUS-CPS included three samples collected in July 2014, January 2015, and May 2015, whereas the 2018/2019 TUS-CPS included three samples collected in July 2018, January 2019, and May 2019. TUS-CPS participants complete the survey using computer-assisted telephone interviewing (CATI) (about two-thirds of the sample) or computer-assisted personal interviewing (CAPI). Response rates in 2014/2015 and 2018/2019 were 54.1% and 56.2%, respectively [27,28]. TUS-CPS is designed to be representative of the U.S. state and national non-institutionalized, civilian adult population, and employs a complex multistage sampling design. Further information on the 2014/2015 and 2018/2019 TUS-CPS waves is publicly available [29]. We restricted our analyses to self-respondents because previous research has shown that the inclusion of proxy respondents may result in underestimation of the prevalence of tobacco use [30].

NHIS is an annual cross-sectional survey designed by the Centers for Disease Control and Prevention’s National Center for Health Statistics to collect information on health status and behaviors, including tobacco use, from U.S. adults aged 18 years or older. Data are collected via personal household interviews using CAPI. Response rates in 2015, 2016, 2017, 2018, and 2019 were 55.2%, 54.3%, 53.0%, 53.1%, and 59.1%, respectively [31,32,33,34,35]. NHIS is designed to be nationally representative of the U.S. non-institutionalized, civilian adult population and employs a complex multistage sampling design. Further details on the methodology of NHIS are published online [36]. We did not include the 2014 NHIS because respondents were not asked questions about the frequency (every day or some days) of cigar/cigarillos/little filtered cigar and traditional pipe use.

Pooling data resulted in a sample of 288,361 adults aged 18 years and over in TUS-CPS (2014/2015 = 157,535 and 2018/2019 = 130,826) and 150,856 adults aged 18 years and over in NHIS (2015 = 33,672, 2016 = 33,028, 2017 = 26,742, 2018 = 25,417, 2019 = 31,997). After excluding missing values for tobacco use variables (TUS = 1.6%, NHIS = 2.1%) and covariates (TUS-CPS = 0.0%, NHIS = 0.4%), the final analytic samples consisted of 283,825 respondents in TUS-CPS and 147,649 respondents in NHIS.

### 2.2. Measures

#### 2.2.1. Tobacco Product Use

We defined four groups of exclusive tobacco product use: (1) cigarette use, (2) ENDS use, (3) OC use, and (4) SLT use. We defined cigarette use as those who had smoked at least 100 cigarettes in their lifetime and now smoked every day or some days. ENDS, OC, and SLT use were defined based only on everyday or some day use. The OC category included the use of cigars/cigarillos/little filtered cigars and traditional pipe/hookah.

In addition to the four exclusive tobacco product use groups, we also defined five dual use and polyuse groups: (1) cigarettes and ENDS, (2) cigarettes and OC, (3) cigarettes and SLT, (4) polyuse with cigarettes (i.e., cigarettes, ENDS, and OC; cigarettes, ENDS, and SLT; cigarettes, OC, and SLT; all four tobacco product groups), and (5) dual/polyuse without cigarettes (i.e., ENDS and OC; ENDS and SLT; OC and SLT; ENDS, OC, and SLT). The nine tobacco use categories were mutually exclusive.

#### 2.2.2. Covariates

We included the following covariates: sex (female, male); age (18–24 years, 25–34 years, 35–54 years, and 55 years or older); race/ethnicity (Hispanic, Non-Hispanic (NH) White, NH Black, another race/ethnicity). The other race/ethnicity group included NH American Indian/Alaska Native, NH Asian/other Pacific Islander, and NH Multiracial adults in NHIS, and American Indian/Alaska Native, Asian Only, Hawaiian/Pacific Island Only and NH Multiracial groups in TUS-CPS. Appendix A displays the prevalence of each covariate by year for TUS-CPS and NHIS from 2014/2015 to 2018/2019.

### 2.3. Statistical Analyses

We estimated the prevalence and trends for use of any tobacco product, the four exclusive tobacco use outcomes, and the five dual use/polyuse outcomes overall and by age, sex, and race/ethnicity. First, we estimated the prevalence and 95% confidence intervals (CI) for each outcome and stratified them by each covariate. Second, we conducted two-sample tests for linear differences in proportions overall and by each covariate to determine the changes in crude prevalence estimates between 2014/2015 and 2018/2019 for TUS-CPS and between 2015 and 2019 for NHIS using the lincom command in Stata. In all analyses, we used sampling weights to account for the complex survey designs and survey specific methods for variance calculation. Specifically, for TUS-CPS, we used balanced repeated replication with replicate weights, with Fay’s adjustment set to 0.4 [37]. For NHIS, we used Taylor series linearization [38]. We conducted all analyses using Stata 16.1 [39].

## 3. Results

### 3.1. Change in Prevalence of Any, Exclusive, Dual and Polytobacco Use over Time

The prevalence of exclusive ENDS use increased from 0.68% in 2014/2015 to 1.22% in 2018/2019 in TUS-CPS and from 1.26% in 2015 to 2.35% in 2019 in NHIS (Table 1). Dual use of cigarettes + ENDS decreased in both TUS-CPS (1.33% to 0.74%) and NHIS (1.62% to 1.25%). Many of the tobacco product use outcomes trended in the same direction in TUS-CPS and NHIS. However, more statistically significant changes over time were observed in TUS-CPS than in NHIS, likely due to sample size limitations in NHIS. For example, the prevalence of exclusive cigarette use, dual use of cigarettes + SLT, and polyuse with cigarettes decreased in TUS-CPS but did not statistically significantly decrease in NHIS over time. In addition, dual/polyuse without cigarettes statistically significantly increased in NHIS (0.37% to 0.63%) but not TUS-CPS (0.25% to 0.29%). Appendix A visualizes the trends in prevalence of any, exclusive, dual, and polytobacco use over time by survey. We display trends in all tobacco product use outcomes overall and by age, sex, and race/ethnicity for NHIS years 2015, 2016, 2017, 2018, and 2019 in Appendix A.

### 3.2. Changes in the Prevalence of Any and Exclusive Tobacco Use over Time by Age

From 2014/2015 to 2018/2019, use of any tobacco product decreased among respondents aged 18–24 (18.53% to 13.86%), 25–34 (20.52% to 17.55%), and 35–54 (19.14% to 17.04%) in TUS-CPS and 18–24 (21.47% to 18.13%) in NHIS (Table 2). Exclusive cigarette use also decreased among the same age groups in both surveys, whereas exclusive OC use and exclusive SLT use decreased among respondents aged 18–24. Exclusive ENDS use increased among all age groups in TUS-CPS and respondents aged 18–24, 25–34, and 35–54 in NHIS. For example, among respondents aged 18–24, exclusive ENDS use increased from 1.12% to 3.29 in TUS-CPS and 2.39% to 5.46% in NHIS.

### 3.3. Changes in the Prevalence of Dual and Polytobacco Use over Time by Age

Dual use of cigarettes + ENDS decreased among all age groups in TUS-CPS and among respondents aged 35–54 and 55+ in NHIS from 2014/2015 to 2018/2019. However, this use pattern increased among respondents aged 25–34 in NHIS (1.48% to 2.19%) (Table 3). Dual use of cigarettes + OC decreased in TUS-CPS among respondents aged 18–24 (1.36% to 0.44%) and 25–34 (1.05% to 0.72%). In addition, dual use of cigarettes + SLT decreased in NHIS among respondents aged 18–24 (0.84% to 0.19%). Polyuse with cigarettes decreased among respondents aged 25–34, 35–54, and 55+ in TUS-CPS, whereas dual/polyuse without cigarettes increased in TUS-CPS among respondents aged 35–54, and among NHIS respondents aged 18–24 and 25–34.

### 3.4. Changes in the Prevalence of Any and Exclusive Tobacco Use over Time by Sex

From 2014/2015 to 2018/2019, the use of any tobacco product decreased among females (13.21% to 11.47%) and males (21.65% to 19.31%) in TUS-CPS but not NHIS (Table 4). Exclusive use of cigarettes, OC, and SLT decreased among males, and exclusive cigarette use decreased among females in TUS-CPS. Exclusive ENDS use increased among females and males in TUS-CPS (0.54% to 1.02% and 0.82% to 1.45%, respectively) and NHIS (0.89% to 1.93% and 1.66% and 2.81%, respectively).

### 3.5. Changes in the Prevalence of Dual and Polytobacco Use over Time by Sex

Dual use of cigarettes + ENDS decreased among females in TUS-CPS (1.40% to 0.67%) and males in TUS-CPS (1.24% to 0.81%) and NHIS (1.73% to 1.27%) from 2014/2015 to 2018/2019 (Table 5). Dual use of cigarettes + OC (1.27% to 0.95%) and cigarettes + SLT (0.43% to 0.35%) decreased among males in TUS-CPS. Polyuse with cigarettes decreased among females (0.14% to 0.07%) and males (0.50% and 0.37%) in TUS-CPS, whereas dual/polyuse without cigarettes increased among males in NHIS (0.68% to 1.10%).

### 3.6. Changes in the Prevalence of Any and Exclusive Tobacco Use over Time by Race/Ethnicity

From 2014/2015 to 2018/2019, the use of any tobacco product decreased among all racial/ethnic groups in TUS-CPS and increased among respondents who identified as another race/ethnicity in NHIS (11.78% to 16.31%) (Table 6). Exclusive cigarette use decreased among all racial/ethnic groups in TUS-CPS and among Hispanic respondents in NHIS. However, exclusive ENDS use increased among all racial/ethnic groups in both surveys except for NH Blacks in TUS-CPS. Exclusive OC (1.81% to 1.59%) and exclusive SLT (1.60% to 1.47%) decreased among NH White respondents in TUS-CPS. In NHIS, exclusive SLT use increased among respondents who identify as another race/ethnicity (0.29% to 1.12%).

### 3.7. Changes in the Prevalence of Dual and Polytobacco Use over Time by Race/Ethnicity

Dual use of cigarettes + ENDS decreased among NH Black, Hispanic, and respondents of other races/ethnicities in TUS-CPS and NH White respondents in NHIS from 2014/2015 to 2018/2019 (Table 7). Dual use of cigarettes + OC decreased among NH White (0.71% to 0.44%) and NH Black (0.76% to 0.58%) respondents in TUS-CPS and NH Black respondents in NHIS (2.37% to 1.26%). However, in NHIS, dual use of cigarettes + OC increased among another race/ethnicity (0.39% to 0.93%) and Hispanic (0.49% to 0.90%) respondents. Dual use of cigarettes + SLT decreased among Hispanic respondents and polyuse with cigarettes decreased among NH Black respondents in TUS-CPS. Furthermore, dual/polyuse without cigarettes increased among Hispanic respondents in TUS-CPS and NH White, NH Black, and respondents of other race/ethnicities in NHIS.

## 4. Discussion

Our study examined recent trends in exclusive, dual, and polyuse among U.S. adults using two nationally representative surveys with comparable time points. We observed that patterns of exclusive, dual, and polyuse are changing over time in the U.S. adult population. In addition, variations in trends over time by age, sex, and race/ethnicity suggest that the future monitoring of trends in tobacco product use are needed to address disparities. Decreases in cigarette smoking across groups and increases in ENDS use in some sociodemographic groups have had a major influence on these trends. To the best of our knowledge, our study is among the first to characterize and compare trends in exclusive tobacco use, dual use, and polyuse patterns in separate, nationally representative U.S. adult samples, because most recent investigations have focused on youth [40,41].

We found an increasing trend in exclusive ENDS use and a decreasing trend in the dual use of cigarettes and ENDS in both surveys during the study period. These findings suggest that patterns of ENDS use are changing. In 2014/2015, most ENDS users were dual users with cigarettes, whereas in 2018/2019, most ENDS users were exclusive users. One possible explanation is that the dual use of cigarettes and ENDS could be a transitory stage towards cigarette cessation for some smokers [42]. Moreover, previous studies have reported that young adults are initiating tobacco product use with ENDS over other products, which could also explain the increase in exclusive ENDS use [43,44]. Nationally representative longitudinal studies are necessary to evaluate transitions from dual use involving ENDS to exclusive ENDS use, and vice versa, to understand these relationships more clearly [45].

Our study found that although overall cigarette use has decreased, there are discrepancies in the trends of exclusive cigarette use by survey; exclusive cigarette use statistically significantly decreased over time in TUS-CPS but not in NHIS. To some extent, inter-survey differences may be explained by differences in statistical power, because TUS-CPS has a larger sample than NHIS. In addition, TUS-CPS estimates variance by using the balanced repeated replication method, which produces smaller standard errors and narrower confidence intervals, and which may lead to more precise findings than in NHIS, which uses Taylor linearization series [37,38]. Given the larger sample of TUS-CPS, and the overall decreases in smoking in the United States, it is likely that exclusive cigarette smoking is indeed decreasing, as suggested by TUS-CPS [1]. Our findings demonstrate that most dual use and polyuse groups including cigarettes are also decreasing. Previous studies examining tobacco use trends have described cigarette use overall and not stratified it into exclusive, dual, and poly-cigarette users [8,9]. Our analyses can help capture changes in patterns of cigarette smoking and tobacco product use over time more granularly, especially in the context of using multiple tobacco products.

In some cases, we observed variations in trends between surveys. For example, dual/polyuse without cigarettes increased in NHIS but not TUS-CPS. Additionally, dual use and polyuse with cigarette groups decreased in TUS-CPS but not NHIS, possibly due to statistical power issues. Survey modality might additionally explain differences in results between TUS-CPS and NHIS. In TUS-CPS, more than 60% of respondents answered surveys via telephone interviews [27,28], whereas all respondents in NHIS answered surveys via in-person interviews [31,32,33,34,35]. Research has indicated that some populations might under-report their smoking status over the phone, possibly making it more difficult to compare results between surveys with different methodologies [30]. Another possible explanation is the ways in which tobacco use questions were asked between surveys. For example, respondents in TUS-CPS were asked about whether they smoked traditional pipes and hookah separately, whereas respondents in NHIS were asked about whether they smoked pipes and hookah in one question. However, these minor differences likely did not affect our estimates, given that we examined the use of cigars/cigarillos/little filtered cigars and traditional pipe/hookah as one group. Despite minor methodological differences between surveys, a more thorough investigation of inter-survey variations is needed to better understand these differences in findings. Future research aimed at understanding the impact of using different survey methodologies to estimate the prevalence of dual and polytobacco use may benefit tobacco control researchers and policymakers.

Our study comes with limitations. Sample sizes, especially in NHIS, precluded our ability to examine more granular tobacco product use combinations. Collapsing dual use and polyuse groups into those with and without cigarettes was performed to differentiate tobacco users who use cigarettes from those who do not, under the assumption that health effects differ between these groups. Nevertheless, future studies should attempt to disaggregate these categories to better understand trends in specific dual use and polyuse groups. Our analysis used repeated cross-sectional data to understand population-level trends in prevalence; examining transitions in tobacco product use for individuals requires longitudinal studies that ask about the use of multiple tobacco products over time. Despite these limitations, our study is one of the first to characterize population-level prevalence and trends in exclusive, dual, and polytobacco use among U.S. adults using data from two nationally representative surveys.

## 5. Conclusions

Our findings demonstrate that exclusive ENDS use increased over time, whereas the dual use of cigarettes and ENDS decreased over time in two nationally representative surveys. Results also suggest that polyuse with cigarettes is decreasing over time, whereas dual/polyuse without cigarettes is increasing. These findings from two nationally representative surveys show how patterns of exclusive, dual, and polyuse are changing rapidly in the current complex context of widely available tobacco products, with variation in trends by age, sex, and race/ethnicity. Continued monitoring of dual use and polyuse is needed to better contextualize population-level use patterns, differences in use patterns by key sociodemographic groups, and inform prevention and cessation efforts that address the use of multiple tobacco products.

## Figures and Tables

**Table 1 ijerph-18-13092-t001:** Population prevalence of exclusive, dual and polytobacco use among U.S. adults from 2014/2015 to 2018/2019.

Tobacco Product Use	TUS-CPS					NHIS				
	2014/2015		2018/2019		*p*-Value	2015		2019		*p*-Value
	%	95% CI	%	95% CI		%	95% CI	%	95% CI	
Any tobacco product	17.27	17.09–17.46	15.25	15.03–15.47	**<0.001**	20.33	19.66–21.00	20.80	20.10–21.40	0.367
Exclusive cigarette	10.89	10.75–11.03	9.43	9.25–9.60	**<0.001**	11.36	10.86–11.86	10.85	10.38–11.33	0.143
Exclusive ENDS	0.68	0.64–0.72	1.22	1.16–1.29	**<0.001**	1.26	1.09–1.43	2.35	2.14–2.58	**<0.001**
Exclusive OC	1.65	1.58–1.72	1.55	1.48–1.62	0.064	2.33	2.07–2.58	2.23	2.02–2.46	0.552
Exclusive SLT	1.17	1.11–1.22	1.03	0.97–1.09	**0.001**	1.41	1.19–1.62	1.57	1.40–1.77	0.246
Dual cigarettes + ENDS	1.33	1.27–1.39	0.74	0.60–0.79	**<0.001**	1.62	1.42–1.81	1.25	1.10–1.43	**0.005**
Dual cigarettes + OC	0.79	0.74–0.84	0.60	0.56–0.65	**<0.001**	1.09	0.94–1.24	1.10	0.97–1.24	0.950
Dual cigarettes + SLT	0.22	0.20–0.24	0.18	0.16–0.20	**<0.001**	0.37	0.26–0.47	0.30	0.23–0.38	0.287
Poly with cigarettes	0.31	0.28–0.34	0.21	0.18–0.25	**<0.001**	0.53	0.41–0.64	0.48	0.38–0.60	0.554
Dual/Poly w/o cigarettes	0.25	0.25–0.27	0.29	0.26–0.33	0.059	0.37	0.27–0.47	0.63	0.51–0.77	**0.002**

TUS-CPS: Tobacco Use Supplement to the Current Population Survey; NHIS: National Health Interview Survey; ENDS: electronic nicotine delivery systems; OC: other combustible tobacco products; SLT: smokeless tobacco products. Bolded text indicates statistical significance (*p* < 0.05).

**Table 2 ijerph-18-13092-t002:** Population prevalence of any and exclusive tobacco use among U.S. adults from 2014/2015 to 2018/2019 by age group.

Tobacco Product Use	TUS-CPS					NHIS				
	2014/2015		2018/2019		*p*-Value	2015		2019		*p*-Value
	%	95% CI	%	95% CI		%	95% CI	%	95% CI	
Any tobacco product										
18–24	18.53	17.85–19.23	13.86	13.11–14.65	**<0.001**	21.47	19.40–23.69	18.13	16.25–20.17	**0.024**
25–34	20.52	20.02–21.03	17.55	17.05–18.07	**<0.001**	24.40	22.78–26.11	25.85	24.28–27.48	0.218
35–54	19.14	18.83–19.45	17.04	16.70–17.38	**<0.001**	23.37	22.20–24.58	24.54	23.43–25.68	0.162
55+	13.36	13.11–13.62	13.02	12.73–13.31	0.053	15.07	14.25–15.93	15.94	15.19–16.72	0.135
Exclusive cigarette										
18–24	9.12	8.57–9.71	5.21	4.75–5.72	**<0.001**	8.43	7.10–9.99	4.51	3.66–5.56	**<0.001**
25–34	12.24	11.85–12.65	9.55	9.18–9.93	**<0.001**	13.15	12.07–14.30	11.91	10.78–13.13	0.133
35–54	12.54	12.29–12.81	11.12	10.81–11.45	**<0.001**	13.55	12.63–14.53	13.09	12.22–14.01	0.488
55+	9.25	9.04–9.47	9.26	9.03–9.50	0.932	9.44	8.78–10.14	10.41	9.79–11.06	**0.041**
Exclusive ENDS										
18–24	1.12	0.93–1.34	3.29	2.92–3.70	**<0.001**	2.39	1.73–3.30	5.46	4.47–6.66	**<0.001**
25–34	0.91	0.80–1.03	1.80	1.62–1.99	**<0.001**	1.56	1.19–2.05	3.57	2.97–4.28	**<0.001**
35–54	0.71	0.65–0.79	0.98	0.88–1.09	**<0.001**	1.35	1.10–1.66	2.37	2.05–2.75	**<0.001**
55+	0.37	0.32–0.41	0.48	0.43–0.54	**0.002**	0.64	0.48–0.85	0.80	0.65–0.98	0.201
Exclusive OC										
18–24	2.55	2.27–2.87	1.58	1.31–1.89	**<0.001**	4.07	3.20–5.17	1.85	1.28–2.65	**<0.001**
25–34	2.18	2.01–2.37	2.15	1.95–2.37	0.819	3.02	2.41–3.78	2.62	2.00–3.41	0.418
35–54	1.35	1.26–1.45	1.51	1.39–1.64	**0.048**	2.06	1.66–2.55	2.76	2.37–3.21	**0.023**
55+	1.34	1.25–1.43	1.28	1.20–1.37	0.379	1.64	1.36–1.98	1.70	1.47–1.98	0.766
Exclusive SLT										
18–24	1.17	1.00–1.36	0.91	0.74–1.12	0.069	1.49	0.98–2.25	1.14	0.69–1.90	0.423
25–34	1.26	1.13–1.41	1.12	1.00–1.27	0.142	1.54	1.15–2.06	1.73	1.30–2.31	0.568
35–54	1.53	1.44–1.63	1.30	1.21–1.41	**0.001**	1.71	1.38–2.13	2.10	1.79–2.45	0.131
55+	0.76	0.70–0.82	0.78	0.72–0.86	0.584	1.02	0.76–1.38	1.18	0.97–1.44	0.413

TUS-CPS: Tobacco Use Supplement to the Current Population Survey; NHIS: National Health Interview Survey; ENDS: electronic nicotine delivery systems; OC: other combustible tobacco products; SLT: smokeless tobacco products. Bolded text indicates statistical significance (*p* < 0.05).

**Table 3 ijerph-18-13092-t003:** Population prevalence of dual and polytobacco use among U.S. adults from 2014/2015 to 2018/2019 by age group.

Tobacco Product Use	TUS-CPS					NHIS				
	2014/2015		2018/2019		*p*-Value	2015		2019		*p*-Value
	%	95% CI	%	95% CI		%	95% CI	%	95% CI	
Dual cigarettes + ENDS										
18–24	1.30	1.10–1.53	0.81	0.63–1.03	**0.002**	1.53	1.10–2.14	1.35	0.87–2.08	0.636
25–34	1.68	1.53–1.84	1.16	1.03–1.30	**<0.001**	1.48	1.10–1.98	2.19	1.74–2.74	**0.035**
35–54	1.58	1.49–1.68	0.80	0.72–0.89	**<0.001**	2.26	1.92–2.65	1.60	1.34–1.92	**0.006**
55+	0.91	0.84–0.98	0.46	0.40–0.52	**<0.001**	1.11	0.88–1.39	0.48	0.38–0.62	**<0.001**
Dual cigarettes + OC										
18–24	1.36	1.15–1.60	0.44	0.33–0.59	**<0.001**	0.85	0.52–1.40	1.12	0.72–1.71	0.419
25–34	1.05	0.92–1.20	0.72	0.63–0.84	**<0.001**	1.61	1.22–2.13	1.16	0.83–1.61	0.131
35–54	0.76	0.69–0.83	0.68	0.62–0.76	0.185	1.24	0.99–1.56	1.24	1.01–1.51	0.992
55+	0.47	0.41–0.53	0.52	0.47–0.58	0.209	0.78	0.59–1.02	0.94	0.78–1.14	0.242
Dual cigarettes + SLT										
18–24	0.37	0.29–0.48	0.23	0.14–0.37	0.094	0.84	0.42–1.67	0.19	0.07–0.53	**0.037**
25–34	0.37	0.31–0.43	0.28	0.22–0.36	0.060	0.57	0.34–0.96	0.48	0.31–0.74	0.626
35–54	0.25	0.21–0.29	0.22	0.18–0.27	0.313	0.31	0.21–0.48	0.42	0.29–0.60	0.307
55+	0.07	0.05–0.09	0.07	0.05–0.09	0.844	0.15	0.08–0.29	0.14	0.09–0.23	0.871
Poly with cigarettes										
18–24	0.80	0.65–1.00	0.59	0.43–0.80	0.120	1.04	0.64–1.71	0.85	0.47–1.54	0.595
25–34	0.48	0.40–0.58	0.34	0.26–0.44	**0.019**	1.03	0.68–1.54	1.03	0.72–1.45	1.000
35–54	0.25	0.21–0.29	0.19	0.16–0.23	**0.022**	0.45	0.32–0.64	0.48	0.35–0.66	0.794
55+	0.11	0.09–0.13	0.05	0.04–0.07	**0.001**	0.18	0.10–0.30	0.10	0.06–0.20	0.231
Dual/Poly w/o cigarettes										
18–24	0.74	0.61–0.89	0.81	0.63–1.04	0.588	0.81	0.48–1.36	1.66	1.11–2.46	**0.033**
25–34	0.35	0.29–0.43	0.43	0.35–0.54	0.138	0.45	0.27–0.77	1.18	0.86–1.60	**0.001**
35–54	0.16	0.13–0.20	0.23	0.18–0.28	**0.023**	0.44	0.28–0.69	0.48	0.33–0.68	0.775
55+	0.10	0.07–0.13	0.10	0.08–0.13	0.623	0.12	0.06–0.24	0.18	0.10–0.30	0.347

TUS-CPS: Tobacco Use Supplement to the Current Population Survey; NHIS: National Health Interview Survey; ENDS: electronic nicotine delivery systems; OC: other combustible tobacco products; SLT: smokeless tobacco products. Bolded text indicates statistical significance (*p* < 0.05).

**Table 4 ijerph-18-13092-t004:** Population prevalence of any and exclusive tobacco use among U.S. adults from 2014/2015 to 2018/2019 by sex.

Tobacco Product Use	TUS-CPS					NHIS				
	2014/2015		2018/2019		*p*-Value	2015		2019		*p*-Value
	%	95% CI	%	95% CI		%	95% CI	%	95% CI	
Any tobacco product										
Female	13.21	12.99–13.44	11.47	11.23–11.72	**<0.001**	15.38	14.65–16.13	15.69	14.95–16.46	0.560
Male	21.65	21.38–19.31	19.31	18.96–19.67	**<0.001**	25.64	24.56–26.75	26.15	25.24–27.09	0.481
Exclusive cigarette										
Female	10.11	9.92–10.30	8.76	8.55–8.97	**<0.001**	11.20	10.58–11.85	10.76	10.16–11.39	0.325
Male	11.73	11.52–11.94	10.15	9.89–10.41	**<0.001**	11.53	10.84–12.26	10.94	10.29–11.62	0.233
Exclusive ENDS										
Female	0.54	0.50–0.59	1.02	0.94–1.10	**<0.001**	0.89	0.72–1.11	1.93	1.66–2.23	**<0.001**
Male	0.82	0.76–0.89	1.45	1.35–1.56	**<0.001**	1.66	1.40–1.96	2.81	2.49–3.16	**<0.001**
Exclusive OC										
Female	0.51	0.46–0.57	0.51	0.46–0.57	0.932	0.84	0.65–1.09	0.69	0.53–0.88	0.253
Male	2.88	2.75–2.81	2.67	2.53–2.81	**0.034**	3.92	3.49–4.40	3.87	3.47–4.32	0.880
Exclusive SLT										
Female	0.09	0.08–0.11	0.08	0.07–0.10	0.511	0.13	0.09–0.21	0.17	0.10–0.27	0.538
Male	2.32	2.22–2.44	2.05	1.93–2.18	**0.001**	2.77	2.37–3.24	3.07	2.73–3.47	0.298

TUS-CPS: Tobacco Use Supplement to the Current Population Survey; NHIS: National Health Interview Survey; ENDS: electronic nicotine delivery systems; OC: other combustible tobacco products; SLT: smokeless tobacco products. Bolded text indicates statistical significance (*p* < 0.05).

**Table 5 ijerph-18-13092-t005:** Population prevalence of dual and polytobacco use among U.S. adults from 2014/2015 to 2018/2019 by sex.

Tobacco Product Use	TUS-CPS					NHIS				
	2014/2015		2018/2019		*p*-Value	2015		2019		*p*-Value
	%	95% CI	%	95% CI		%	95% CI	%	95% CI	
Dual cigarettes + ENDS										
Female	1.40	1.32–1.49	0.67	0.62–0.73	**<0.001**	1.51	1.29–1.78	1.23	1.04–1.46	0.085
Male	1.24	1.16–1.32	0.81	0.74–0.89	**<0.001**	1.73	1.46–2.04	1.27	1.05–1.54	**0.018**
Dual cigarettes + OC										
Female	0.34	0.30–0.38	0.28	0.24–0.32	**0.032**	0.50	0.37–0.66	0.53	0.42–0.67	0.739
Male	1.27	1.18–1.36	0.95	0.87–1.03	**<0.001**	1.72	1.47–2.02	1.70	1.46–1.98	0.901
Dual cigarettes + SLT										
Female	0.03	0.01–0.04	0.02	0.00–0.03	0.169	0.02	0.01–0.06	0.05	0.02–0.10	0.354
Male	0.43	0.39–0.48	0.35	0.30–0.40	**0.009**	0.73	0.55–0.98	0.57	0.44–0.73	0.201
Poly with cigarettes										
Female	0.14	0.11–0.17	0.07	0.05–0.10	**<0.001**	0.18	0.10–0.32	0.16	0.10–0.26	0.731
Male	0.50	0.45–0.56	0.37	0.32–0.43	**0.003**	0.90	0.71–1.14	0.82	0.64–1.06	0.618
Dual/Poly w/o cigarettes										
Female	0.05	0.04–0.07	0.08	0.06–0.11	0.067	0.09	0.05–0.17	0.19	0.11–0.31	0.075
Male	0.46	0.41–0.51	0.52	0.45–0.59	0.162	0.68	0.51–0.91	1.10	0.88–1.36	**0.008**

TUS-CPS: Tobacco Use Supplement to the Current Population Survey; NHIS: National Health Interview Survey; ENDS: electronic nicotine delivery systems; OC: other combustible tobacco products; SLT: smokeless tobacco products. Bolded text indicates statistical significance (*p* < 0.05).

**Table 6 ijerph-18-13092-t006:** Population prevalence of any and exclusive tobacco use among U.S. adults from 2014/2015 to 2018/2019 by race/ethnicity.

Tobacco Product Use	TUS-CPS					NHIS				
	2014/2015		2018/2019		*p*-Value	2015		2019		*p*-Value
	%	95% CI	%	95% CI		%	95% CI	%	95% CI	
Any tobacco product										
Hispanic	10.87	10.42–11.33	9.54	9.06–10.05	**<0.001**	13.03	11.88–14.28	13.14	11.91–14.46	0.909
NH White	19.39	19.15–19.64	17.24	16.94–17.54	**<0.001**	22.81	21.91–23.74	23.29	22.52–24.07	0.433
NH Black	17.06	16.41–17.72	15.56	14.93–16.22	**0.001**	21.20	19.46–23.05	20.70	19.02–22.49	0.691
Another race/ethnicity	12.84	12.25–13.46	11.32	10.76–11.91	**0.001**	11.78	10.12–13.66	16.31	14.39–18.44	**0.001**
Exclusive cigarette										
Hispanic	7.53	7.16–7.90	6.23	5.85–6.63	**<0.001**	8.50	7.57–9.52	6.91	6.03–7.90	**0.021**
NH White	11.78	11.60–11.98	10.34	10.11–10.56	**<0.001**	12.27	11.61–12.95	11.86	11.29–12.45	0.367
NH Black	12.07	11.52–12.64	10.74	10.19–11.32	**0.001**	12.77	11.51–14.14	12.34	10.98–13.83	0.662
Another race/ethnicity	8.43	7.95–8.94	7.16	6.67–7.67	**0.001**	6.84	5.67–8.24	8.71	7.25–10.43	0.072
Exclusive ENDS										
Hispanic	0.43	0.34–0.54	0.75	0.62–0.91	**<0.001**	0.95	0.63–1.44	1.72	1.32–2.24	**0.012**
NH White	0.82	0.76–0.88	1.52	1.44–1.63	**<0.001**	1.50	1.29–1.74	2.63	2.37–2.92	**<0.001**
NH Black	0.33	0.25–0.45	0.44	0.33–0.60	0.168	0.57	0.36–0.90	1.63	1.12–2.35	**0.002**
Another race/ethnicity	0.52	0.41–0.67	0.98	0.80–1.21	**<0.001**	0.90	0.49–1.65	2.41	1.67–3.45	**0.004**
Exclusive OC										
Hispanic	1.07	0.92–1.24	1.10	0.93–1.30	0.807	1.72	1.30–2.28	2.16	1.55–3.01	0.319
NH White	1.81	1.73–1.89	1.59	1.50–1.68	**0.001**	2.39	2.08–2.74	2.20	1.96–2.47	0.358
NH Black	2.04	1.81–2.30	2.39	2.11–2.70	0.100	3.36	2.61–4.32	3.23	2.57–4.04	0.811
Another race/ethnicity	0.93	0.74–1.17	0.96	0.76–1.21	0.867	1.34	0.85–2.11	1.23	0.81–1.85	0.785
Exclusive SLT										
Hispanic	0.22	0.16–0.30	0.15	0.10–0.21	0.079	0.26	0.12–0.53	0.19	0.10–0.35	0.526
NH White	1.60	1.53–1.68	1.47	1.38–1.57	**0.033**	1.97	1.67–2.31	2.20	1.95–2.48	0.260
NH Black	0.40	0.31–0.49	0.28	0.20–0.38	0.058	0.46	0.31–0.70	0.39	0.24–0.63	0.589
Another race/ethnicity	0.58	0.46–0.73	0.54	0.41–0.71	0.700	0.29	0.13–0.67	1.12	0.65–1.93	**0.013**

TUS–CPS: Tobacco Use Supplement to the Current Population Survey; NHIS: National Health Interview Survey; ENDS: electronic nicotine delivery systems; OC: other combustible tobacco products; SLT: smokeless tobacco products. Bolded text indicates statistical significance (*p* < 0.05).

**Table 7 ijerph-18-13092-t007:** Population prevalence of dual and polytobacco use among U.S. adults from 2014/2015 to 2018/2019 by race/ethnicity.

Tobacco Product Use	TUS-CPS					NHIS				
	2014/2015		2018/2019		*p*-Value	2015		2019		*p*-Value
	%	95% CI	%	95% CI		%	95% CI	%	95% CI	
Dual cigarettes + ENDS										
Hispanic	1.21	1.02–1.43	0.65	0.50–0.84	**<0.001**	0.69	0.49–0.98	0.59	0.37–0.93	0.570
NH White	0.52	0.43–0.64	0.43	0.33–0.56	0.211	1.93	1.67–2.23	1.49	1.27–1.73	**0.015**
NH Black	1.65	1.57–1.73	0.90	0.84–0.97	**<0.001**	1.28	0.87–1.86	1.04	0.67–1.63	0.502
Another race/ethnicity	0.66	0.55–0.80	0.35	0.25–0.49	**0.001**	1.34	0.81–2.22	1.07	0.64–1.79	0.540
Dual cigarettes + OC										
Hispanic	0.56	0.43–0.74	0.50	0.39–0.64	0.545	0.49	0.33–0.72	0.90	0.62–1.30	**0.036**
NH White	0.71	0.59–0.86	0.44	0.35–0.56	**0.004**	1.08	0.91–1.27	1.14	0.98–1.33	0.610
NH Black	0.76	0.70–0.82	0.58	0.52–0.63	**<0.001**	2.37	1.85–3.03	1.26	0.88–1.79	**0.003**
Another race/ethnicity	1.20	1.02–1.40	1.03	0.87–1.22	0.202	0.39	0.20–0.73	0.93	0.60–1.43	**0.024**
Dual cigarettes + SLT										
Hispanic	0.14	0.10–0.21	0.06	0.03–0.10	**0.020**	0.04	0.01–0.13	0.13	0.05–0.35	0.225
NH White	0.04	0.02–0.08	0.05	0.03–0.10	0.556	0.53	0.39–0.71	0.42	0.33–0.54	0.285
NH Black	0.30	0.27–0.34	0.25	0.22–0.29	**0.029**	0.09	0.03–0.27	0.04	0.01–0.15	0.389
Another race/ethnicity	0.06	0.03–0.10	0.03	0.01–0.08	0.316	0.05	0.02–0.15	0.04	0.01–0.15	0.738
Poly with cigarettes										
Hispanic	0.31	0.21–0.45	0.17	0.09–0.29	0.098	0.24	0.12–0.51	0.27	0.13–0.53	0.856
NH White	0.20	0.14–0.29	0.19	0.13–0.30	0.908	0.66	0.51–0.84	0.60	0.46–0.78	0.600
NH Black	0.36	0.33–0.40	0.24	0.21–0.29	**<0.001**	0.16	0.08–0.31	0.20	0.10–0.42	0.654
Another race/ethnicity	0.17	0.12–0.24	0.11	0.06–0.20	0.212	0.58	0.17–1.92	0.40	0.20–0.83	0.651
Dual/Poly w/o cigarettes										
Hispanic	0.16	0.09–0.27	0.30	0.21–0.43	**0.045**	0.14	0.05–0.35	0.27	0.14–0.55	0.242
NH White	0.16	0.10–0.23	0.19	0.14–0.27	0.421	0.50	0.38–0.67	0.75	0.60–0.94	**0.027**
NH Black	0.30	0.26–0.34	0.33	0.29–0.38	0.282	0.15	0.06–0.37	0.58	0.30–1.11	**0.035**
Another race/ethnicity	0.14	0.09–0.21	0.19	0.12–0.30	0.298	0.05	0.01–0.18	0.40	0.18–0.89	**0.033**

TUS-CPS: Tobacco Use Supplement to the Current Population Survey; NHIS: National Health Interview Survey; ENDS: electronic nicotine delivery systems; OC: other combustible tobacco products; SLT: smokeless tobacco products. Bolded text indicates statistical significance (*p* < 0.05).

## Data Availability

Publicly available datasets were analyzed in this study. TUS-CPS data presented in this study are openly available by the NIH Division of Cancer Control & Population Sciences at https://cancercontrol.cancer.gov/brp/tcrb/tus-cps, accessed on 16 November 2021 and NHIS data are available by the CDC at https://www.cdc.gov/nchs/nhis/index.htm, accessed on 16 November 2021 or by IPUMS at https://nhis.ipums.org/nhis/, accessed on 16 November 2021.

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
