# Peer review of "Trends in Exclusive, Dual and Polytobacco Use among U.S. Adults, 2014–2019: Results from Two Nationally Representative Surveys"

_ijerph, 2021, doi:10.3390/ijerph182413092_

Round 1
Reviewer 1 Report
This manuscript makes a meaningful contribution to the literature using two robust datasets regarding recent changing patterns of tobacco use. The manuscript is very well written. I have only a few minor comments for the authors to consider in a potential revision.
- Include a paragraph summarizing the methodology of both the CPS-TUS and the NHIS. Such as study design (e.g., TUS is designed to yield state level estimates—hence the large sample size), modes of data collection, response rate, etc..
- Is possible to also examine menthol cigarette use? Both datasets have measures on menthol. At a minimum – this would be interesting to include in Table 1- where Exclusive cigarette smoking- could be broken down into menthol and non-menthol exclusive smoking. In this regard, consider the take away from Ref #9 – which shows that much of the declines in cigarette smoking are attributed to non-menthol.
- Consider inclusion of Weinberger et al., “Trends in Cigar Use in the United States, 2002-2016: Diverging Trends by Race/Ethnicity”
- To facilitate easier comprehension of the tables- consider including an up or down arrow next to significant p values.
- Can the authors comment in the discussion on differing or conflicting patterns across the two datasets? E.g., In table 7- Dual cigs/ENDS- was highest in Hispanic and Black and lowest in whites in the TUS- but in the NHIS, its highest in whites and lowest in Hispanics. Is it possible this is attributed to differences in TUS & NHIS? Are any of these groups over or under-represented in the weighted data differently?
Author Response
Reviewer #1: This manuscript makes a meaningful contribution to the literature using two robust datasets regarding recent changing patterns of tobacco use. The manuscript is very well written. I have only a few minor comments for the authors to consider in a potential revision.
Response: We thank you for your kind words about our manuscript.
- Include a paragraph summarizing the methodology of both the CPS-TUS and the NHIS. Such as study design (e.g., TUS is designed to yield state level estimates—hence the large sample size), modes of data collection, response rate, etc.
Response: We have included this information in the 2.1 Design subsection of the Methods in addition to other information that was already written.
The new text reads, “TUS-CPS is a cross-sectional survey of tobacco use behaviors among adult respondents or proxies aged 18 years or older administered every 3-4 years by the U.S. Census Bureau. The 2014/15 TUS-CPS included three samples collected in July 2014, January 2015, and May 2015, while the 2018/19 TUS-CPS included three samples collected in July 2018, January 2019, and May 2019. TUS-CPS participants complete the survey using Computer Assisted Telephone Interviewing (CATI) (about two-thirds of the sample) or Computer Assisted Personal Interviewing (CAPI). Response rates in 2014/15 and 2018/19 were 54.1% and 56.2%, respectively [27-28]. TUS-CPS is designed to be representative of the U.S. state and national non-institutionalized, civilian adult population and employs a complex multistage sampling design. Further information on the 2014/15 and 2018/19 TUS-CPS waves is publicly available [29]. We restricted our analyses to self-respondents because previous research has shown that the inclusion of proxy respondents may result in underestimation of the prevalence of tobacco use [30].
NHIS is an annual cross-sectional survey designed by the Centers for Disease Control and Prevention’s National Center for Health Statistics to collect information on health status and behaviors, including tobacco use, from U.S. adults aged 18 years or older. Data are collected via personal household interviews using CAPI. Response rates in 2015, 2016, 2017, 2018, and 2019 were 55.2, 54.3%, 53.0%, 53.1%, and 59.1%, respectively [31-35]. NHIS is designed to be nationally representative of the U.S. non-institutionalized, civilian adult population and employs a complex multistage sampling design. Further details on the methodology of NHIS are published online [36]. We did not include 2014 NHIS since respondents were not asked questions about frequency (every day or someday) of cigar/cigarillos/little filtered cigar and traditional pipe use.
- Is possible to also examine menthol cigarette use? Both datasets have measures on menthol. At a minimum – this would be interesting to include in Table 1- where Exclusive cigarette smoking- could be broken down into menthol and non-menthol exclusive smoking. In this regard, consider the take away from Ref #9 – which shows that much of the declines in cigarette smoking are attributed to non-menthol.
Response: Thank you for the suggestion. Unfortunately, further stratifying cigarette use by menthol flavoring in the context of our exclusive, dual, and poly use groups that include ENDS, other combustible products, and smokeless tobacco would result in many categories with small cell sizes. However, a recent study from our team explored this important topic in the context of cigarette and ENDS use (citation below).
Usidame B, Hirschtick J, Zavala-Arciniega L, Mattingly DT, Patel A, Meza R, et al. Exclusive and dual menthol / non-menthol cigarette use with ENDS among adults , 2013 – 2019. Prev Med Reports [Internet]. 2021;24:101566. Available from: https://doi.org/10.1016/j.pmedr.2021.101566
- Consider inclusion of Weinberger et al., “Trends in Cigar Use in the United States, 2002-2016: Diverging Trends by Race/Ethnicity”
Response: We have included this reference in the third paragraph of the Introduction with a sentence that reads, “One study using data from the National Health Interview Survey (NHIS) reported that the prevalence of current cigar use has remained relatively stable from 2005 to 2015 (2.3%), while another study using data from the National Surveys on Drug Use and Health (NSDUH) reported that current cigar use has decreased from 2002 (7.9%) to 2016 (6.3%).”
- To facilitate easier comprehension of the tables- consider including an up or down arrow next to significant p values.
Response: Thank you for the suggestion on making it easier to digest the tables. Instead of adding arrows, we bolded statistically significant p-values (p<0.05) and added a footnote to each table indicating what bolded p-values mean.
- Can the authors comment in the discussion on differing or conflicting patterns across the two datasets? E.g., In table 7- Dual cigs/ENDS- was highest in Hispanic and Black and lowest in whites in the TUS- but in the NHIS, its highest in whites and lowest in Hispanics. Is it possible this is attributed to differences in TUS & NHIS? Are any of these groups over or under-represented in the weighted data differently?
Response: In the paragraph before the limitations in our Discussion section, we explain some potential reasons on conflicting tobacco use patterns across the two datasets, such as differences in survey modality, response rates, etc.
To answer your second question, the racial/ethnic groups in both TUS-CPS and NHIS should not be over or underrepresented since the weights are meant to adjust estimates to be nationally representative. Please refer to Table S1 in our supplementary materials for a breakdown of the overall prevalence of race/ethnicity by survey year in both TUS-CPS and NHIS.
Reviewer 2 Report
This is an important topic and I am pleased to see good quality research being done on the changing use of different kinds of tobacco products in adults. The methods are sound and data are based on large samples – the results are very informative and clearly presented. I have just a few minor comments to address.
Minor comments
- Please define ENDS at first mention (Introduction)
- Figure 1 is very informative – I suggest you change the colour of the TUS line as it is hard to see next to the strong colour of the NIHS line
- In figure 1 please define population prevalence (per how many of the population? is this %?)
- Table 2 – there is an error in the title (remove the word Figure)
- Tables 1, 2, 3, 4 and 5 – it is somewhat difficult to pick out the percentages – suggest to put the lower and upper bounds in a single column eg. 17.09-17.46 to make them distinct from the %s which will then be easier to compare. Are these 95% CI? If so it should be written as such.
- Do the authors believe the reason for significant decrease in the TUS but not in the NIHS survey is purely due to sample size? Are there other differences which might explain this? Were the questions asked differently? This is important to see if there are real differences between the surveys.
- How do these results compare to surveys in other countries? Would be worth adding some comparison of trends to the discussion.
Author Response
Reviewer #2: This is an important topic and I am pleased to see good quality research being done on the changing use of different kinds of tobacco products in adults. The methods are sound and data are based on large samples – the results are very informative and clearly presented. I have just a few minor comments to address.
Response: We are pleased that you find our manuscript to be important and of quality.
- Please define ENDS at first mention (Introduction)
Response: We have defined ENDS at first mention.
- Figure 1 is very informative – I suggest you change the colour of the TUS line as it is hard to see next to the strong colour of the NIHS line
Response: Thanks for the suggestion. We have updated the lines in our figure for each dataset to stronger, contrasting colors.
- In figure 1 please define population prevalence (per how many of the population? is this %?)
Response: We apologize for the confusion. These are percentages. We included this information to the left of y-axes in the Figure. We have also included the population prevalence for sample of each dataset in the figure footnotes to provide a better understanding of sample sizes per year.
- Table 2 – there is an error in the title (remove the word Figure)
Response: We could not locate the error in the title but have double checked our work to ensure that there are no mistakes in the figure and table titles.
- Tables 1, 2, 3, 4 and 5 – it is somewhat difficult to pick out the percentages – suggest to put the lower and upper bounds in a single column eg. 17.09-17.46 to make them distinct from the %s which will then be easier to compare. Are these 95% CI? If so it should be written as such.
Response: We have edited the lower bound and upper bound 95% CI columns into one column (now labeled 95% CI) for tables 1-7 and supplementary tables 2-8.
- Do the authors believe the reason for significant decrease in the TUS but not in the NIHS survey is purely due to sample size? Are there other differences which might explain this? Were the questions asked differently? This is important to see if there are real differences between the surveys.
Response: Yes, we believe that one of the reasons that we observed more statistically significant decreases in TUS-CPS and not NHIS was sample size limitations, especially in the dual/poly use groups. To answer your other questions, we added text to the paragraph in the Discussion commenting on inter-survey differences. The new text reads, “Survey modality might additionally explain differences in results between TUS-CPS and NHIS. In TUS-CPS, more than 60% of respondents took surveys via telephone interviews [27,28], while all respondents in NHIS took surveys via in-person interviews [31-35]. Research has indicated that some populations might underreport their smoking status over the phone, possibly making it more difficult to compare results between surveys with different methodologies [30]. Another possible explanation is the ways in which tobacco use questions were asked between surveys. For example, respondents in TUS-CPS were asked about whether they smoked traditional pipe and hookah separately, whereas respondents in NHIS were asked about whether they smoked pipe and hookah in one question. However, these minor differences likely did not affect our estimates given that we examined use of cigars/cigarillos/little filtered cigars and traditional pipe/hookah as one group.”
- How do these results compare to surveys in other countries? Would be worth adding some comparison of trends to the discussion.
Response: We believe that this is an excellent topic for future research, but beyond the scope of our paper. Given that the tobacco market is evolving worldwide, it is interesting to compare our findings with what is observed in other contexts. Our study is somewhat consistent with a recent systematic review that found that polyuse is relatively common in many countries (reference below). This study also found that tobacco products combinations vary according to geographical region. However, differences in definitions of dual and polyuse make comparisons between countries more complex. Therefore, standardizing definitions of dual/polyuse are needed to better understand differences in trends and patterns of tobacco use in different contexts.
Chen, D.T.; Girvalaki, C.; Mechili, E.A.; Millett, C.; Filippidis, F.T. Global Patterns and Prevalence of Dual and Poly-Tobacco Use: A Systematic Review. Nicotine Tob Res 2021, doi:10.1093/ntr/ntab084.